# Long-Term Stable Cycling of Dendrite-Free Lithium Metal Batteries Using ZIF-90@PP Composite Separator

**DOI:** 10.3390/nano14110975

**Published:** 2024-06-04

**Authors:** Shuilan LYU, Xin Zhang, Sheng Huang, Shuanjin Wang, Min Xiao, Dongmei Han, Yuezhong Meng

**Affiliations:** 1The Key Laboratory of Low-Carbon Chemistry & Energy Conservation of Guangdong Province, State Key Laboratory of Optoelectronic Materials and Technologies, School of Materials Science and Engineering, Sun Yat-sen University, Guangzhou 510275, China; lvshlan@mail2.sysu.edu.cn (S.L.); huangsh47@mail.sysu.edu.cn (S.H.); wangshj@mail.sysu.edu.cn (S.W.); stsxm@mail.sysu.edu.cn (M.X.); 2School of Chemical Engineering and Technology, Sun Yat-sen University, Zhuhai 519082, China; zhangx349@mail2.sysu.edu.cn; 3Institute of Chemistry, Henan Academy of Sciences, Zhengzhou 450000, China; 4College of Chemistry, Zhengzhou University, Zhengzhou 450001, China

**Keywords:** lithium metal batteries, separator, lithium dendrite, deposition, ZIF-90

## Abstract

Lithium metal batteries (LMBs) are anticipated to meet the demand for high energy density, but the growth of lithium dendrites seriously hinders its practical application. Herein, we constructed a kind of composite separator (ZIF-90@PP) consisting of zeolite imidazole framework-90 (ZIF-90) and polypropylene (PP) to promote the uniform deposition of Li^+^ and inhibit the growth of lithium dendrites. The aldehyde groups interacting with TFSI^−^ and the nitrogen-containing negative groups attracting Li^+^ of ZIF-90 can facilitate the dissociation of LiTFSI to release more Li^+^, thus alleviating the influence of space charge near the electrode surface and accelerating the transfer of Li^+^. Not only does the excellent electrolyte wettability of ZIF-90 enhance the electrolyte retention capacity of the separator, but the orderly nano-channels in ZIF-90 also restrict the free migration of anions and homogenize the distribution of Li^+^. Consequently, the functional separator achieves a long-term stable Li plating/stripping cycling for over 780 h at 2 mA cm^−2^. Moreover, an impressive average coulombic efficiency of 98.67% at 0.5 C after 300 cycles is realized by Li || LFP full cells based on ZIF-90@PP with a capacity retention rate of 71.22%. The high-rate and long cycling performance of the modified Li || LFP cells further demonstrates the advantages of the ZIF-90@PP composite separator.

## 1. Introduction

New energy electric vehicles are gradually replacing traditional gasoline-powered vehicles, yet commercial lithium-ion batteries (LIBs) fail to meet the demands for ultra-long battery life in electric vehicles [1]. The lithium metal anode of LMB is considered the “Holy Grail” electrode by researchers due to its highest theoretical specific capacity (3862 mAh g^−1^) and the most negative electrochemical potential (−3.04 V vs. SHE) [2,3,4,5,6,7,8]. As the most promising next-generation batteries, LMBs are anticipated to facilitate the widespread adoption of electric vehicles. However, unfortunately, the high reactivity of the lithium anode causes continuous parasitic reaction with electrolytes, rapidly forming an uneven and unstable solid electrolyte interface (SEI) on the anode surface, leading to uneven current density distribution and heterogeneous deposition of Li^+^. This results in the preferential deposition of Li^+^ on local areas of the lithium metal anode, further inducing the formation of lithium dendrites [9,10]. These dendrites can trigger a series of issues, such as “dead lithium”, volume expansion, and battery short circuits, significantly hindering the commercialization of LMBs. Nonetheless, the formation and growth of lithium dendrites can be effectively inhibited by promoting the transport of Li^+^ and guiding the uniform deposition of Li^+^.

The separator plays an important role in the process of Li^+^ transmission. In recent years, many researchers have employed the strategy of modified separators to promote the uniform deposition of Li^+^. The main materials used for separator modification include carbon-based materials [11,12,13,14], polymers [15,16,17,18,19], transition metal compounds [20,21,22,23,24], and metal organic frameworks (MOFs) [25,26,27,28,29,30,31]. MOFs are rich in open metal sites (OMSs) and can coordinate with anions [32,33,34,35]. Meanwhile, their intrinsic nano-channels can also limit the free migration of anions [4,34,36,37,38,39], thus promoting the Li^+^ transference number. In addition, MOFs with large specific surface areas can adsorb more electrolytes [35,40,41], which is beneficial to provide more Li^+^ transport pathways. Therefore, MOFs are considered to be an ideal material for realizing high-performance functional separators. Although most of the MOF-modified separators are used to suppress the “shuttle effect” of lithium–sulfur batteries (LSBs), some studies have shown that they also play an important role in stabilizing the lithium metal anode. Hao et al. [4] coated MOFs on a polypropylene (PP) separator and obtained a Li^+^ transference number ($t_{{\mathrm{Li}}^{+}}$) of 0.68, which was much higher than that of a commercial PP separator. At the same time, MOFs with uniform porous structure are beneficial for the uniform deposition of Li^+^, thus achieving a highly stable lithium plating/stripping cycle of more than 150 h. Han et al. [5] developed a Li-MOF-coated PP functional separator, which could restrict anion migration by steric hindrance effect and prolong the sand time of Li nucleation. In addition, the Li-MOF coating helped to form a solid electrolyte interface (SEI) layer composed of inorganic LiF and Li_3_N, which further inhibited the growth of dendrites. Chen et al. [25] designed a composite separator PP coated with ZIF-67 to overcome the thermal shrinkage and poor electrolyte wettability problems of polyolefin separators. The ZIF-67@PP separator was composed of well-connected ZIF-67, which achieved the anti-thermal shrinkage and high-temperature resistance. The high hydrophilicity of ZIF-67 improved the wettability of the composite separator, and the cell could store more electrolytes. Thus, the cells based on the PP-ZIF-67-CH_3_OH separator achieved more stable cycling at high temperatures of 55 °C, accompanied by significantly improved coulombic efficiency and capacity retention.

A zeolitic imidazolate framework (ZIF) is a kind of MOF material composed of transition metal and imidazolate. In addition to inheriting the excellent properties of most MOFs, it also features exceptional thermal stability due to its similar structure to natural zeolites [33]. Our previous work demonstrated that the interaction of aldehyde groups in PEO-COF solid electrolyte with TFSI^−^ can effectively increase the $t_{{\mathrm{Li}}^{+}}$ of LMBs [42], but the performance of ZIF-based separators containing aldehyde groups still requires further investigation. Herein, we select the more cost-effective ZIF-90 with aldehyde groups to coat a PP separator for stabilizing the lithium metal anode of LMBs. According to Lewis acid–base theory, TFSI^−^ acts as a Lewis base, while the aldehyde groups in ZIF-90 behave as Lewis acids, endowing ZIF-90 with the ability to coordinate with anions to produce more free Li^+^ [33,35,43]. Meanwhile, the negatively charged groups containing nitrogen can attract Li^+^, thus altering its solvation structure, and even directly removing the molecular shell of the solvent to accelerate the transport of Li^+^. The regular nano-channels of ZIF-90 can further homogenize the flux of Li^+^. In addition, the excellent wettability of ZIF-90 with the electrolyte can enhance the electrolyte absorption of the separator, enabling rapid Li^+^ transfer and stable lithium deposition. The $t_{{\mathrm{Li}}^{+}}$ of the cells assembled with ZIF-90@PP functional separators reaches 0.71. Furthermore, the stable plating/stripping cycle of the Li || Li cells exceeds 780 h. The Li || LFP cells based on the ZIF-90@PP separators possess better rate performance and cycle stability. After 300 cycles at 0.5 °C, their average coulombic efficiency reaches 98.67%, and the capacity retention rate is 71.22%.

## 2. Materials and Methods

### 2.1. Synthesis of ZIF-90

The typical synthesis process of ZIF-90 was conducted as follows. First, 1.9217 g imidazole-2-formaldehyde (>98.0%, Aladdin, Shanghai, China) and 1.4875 g Zn(NO_3_)_2_·6H_2_O (AR, Aladdin, Shanghai, China) were added to 50 mL N-N dimethylformamide (AR, Aladdin, Shanghai, China) solvent and stirred for 4 h at 80 °C. When the solution cooled to room temperature, 50 mL methanol (99.5%, Macklin, Shanghai, China) was quickly poured into it under intense agitation and we continued to stir it for 30 min. Next, it was centrifuged to obtain the solid product, which was then washed with methanol and centrifuged 3 times. Finally, the solid product continued to be stirred in methanol for 24 h and transferred to a vacuum oven at 70 °C for 12 h.

### 2.2. Preparation of ZIF-90@PP Separator

Firstly, ZIF-90 and polyvinyl pyrrolidone (PVP) were added into methanol solvent at a mass ratio of 1:1, and then ultrasonically dispersed for 30 min. PVP was used as a binder and dispersant. Next, the dispersion was evenly coated on the PP separators (Celgard 2500) by vacuum filtration and dried in a vacuum oven at 60 °C for 12 h. Finally, the ZIF-90@PP functional separators were cut into 19 mm diameter discs with a load density of ZIF-90 of about 0.09 mg cm^−2^. To highlight the role of functional separators, the blank PP separators with the same diameter were cropped for comparison.

### 2.3. Preparation of Cathodes

LiFePO_4_ powder (LFP) (Canrd, Guangzhou, China), polyvinylidene fluoride (PVDF) (HSV900, Guangzhou, China), and Super P (TIMCAL, Guangzhou, China) with a mass ratio of 8:1:1 were mixed to make a slurry with N-Methylpyrrolidone (NMP) (AR, Aladdin, Shanghai, China). The slurry was obtained after ball milling for 12 h. Afterwards, we used a 200 μm scraper to spread the slurry evenly on the aluminum foil. Next, the aluminum foil coated with slurry was dried in a blast oven at 60 °C for 3 h and then transferred to a vacuum oven at 100 °C overnight. Finally, the aluminum foil was cut into Φ12 mm wafers, in which the load of LFP was about 1.00 mg cm^−2^.

### 2.4. Preparation of Cells

CR2025 coin cells were adopted and the modification layer of the separator was facing the Li anode. The content of the electrolyte (formulated as 1.0 M LiTFSI in DOL:DME = 1:1 vol% with 2.0% LiNO_3_, Canrd, Guangzhou, China) for each cell was 60 μL. The cathode of the Li || LFP cell was LFP prepared in the laboratory, while the cathode and anode of the Li || Li symmetric cell were composed of lithium sheets. The cathode side of the Li || Cu half-cell was a Φ12 mm Cu sheet coated with ZIF-90. All cells were assembled in a glove box with less than 0.1 ppm of moisture and oxygen.

### 2.5. Material Characterization and Electrochemical Measurements

The morphology and elements of the material were detected by scanning electron microscope (SEM, SU8010, Hitachi, Tokyo, Japan) and energy dispersive spectroscopy (EDS, Nexsa, Thermo Fisher Scientific, Waltham, MA, USA). The structure of the material was analyzed by an X-ray diffractometer (XRD, SmartLab, Rigaku, Tokyo, Japan) under the radiation of Cu Kα. The scanning speed was 5° min^−1^, and the scanning diffraction angle 2θ ranged from 5 to 40°. Attenuated total internal reflectance Fourier transform infrared spectroscopy (ATR-FTIR) measurements were carried out on an FTIR spectrometer (Nicolet 6700-Continuμm, Thermo Fisher Scientific, Waltham, MA, USA). All samples were scanned an average of 64 times within the range of 4000 to 600 cm^−1^, with a resolution of 4 cm^−1^. The N_2_ adsorption–desorption curve was obtained by the automatic surface area analyzer (ASAP 2460, Micromeritics, Shanghai, China) at 77 K after vacuum extraction at 150 °C for 12 h. The thermal properties were measured by thermogravimetric analyzer (TG, TGA/DTA, PerkinElmer, Waltham, MA, USA) in N_2_ atmosphere. The contact angle meter (DSA100S, KRÜSS, Hamburg, Germany) was used to test the wettability of the separators. The electrochemical workstation (CHI 604e, Chen Hua, Shanghai, China) was used to test electrochemical impedance spectroscopy (EIS, 0.1~100,000 Hz). Constant current charge and discharge tests were carried out on the LAND battery system (CT200, Wuhan Lanbo, Wuhan, China) at room temperature.

## 3. Results and Discussion

### 3.1. Characterization of Materials

The XRD patterns of the ZIF-90, PP separator, and ZIF-90@PP composite separator are shown in Figure 1a. The diffraction peaks agree with those reported in the literature for similar materials [36,44,45,46], confirming the successful synthesis of ZIF-90. In addition, the XRD pattern of the ZIF-90@PP composite separator displays distinct diffraction peaks corresponding to both ZIF-90 and PP, proving that ZIF-90 is coated onto the PP separator and the crystal structure is not damaged during the coating process. The ATR-FTIR spectra of the ZIF-90, PP, and ZIF-90@PP separator were attained to investigate the characteristic functional groups of the separator (Figure 1b). The presence of aldehyde groups in ZIF-90 is confirmed by the double peaks at 2920 and 2850 cm⁻^1^ and a strong band at 1667 cm⁻^1^, consistent with the literature [44,46]. The broad peak at 3480 cm⁻^1^ is attributed to free –OH stretching vibrations, possibly due to physically adsorbed water from the air [47]. The peaks at 1457, 1420, and 1365 cm⁻^1^ are mainly associated with C–H bending vibrations in PP and ZIF-90. In addition, the peak at 1258 cm^−1^ is attributed to C–N stretching vibrations. The characteristic absorption peaks of the ZIF-90@PP separator match those of both PP and ZIF-90, further indicating successful coating of ZIF-90 onto the PP separator.

The thermal stability of the separator is crucial to the safety of the battery. Figure 1c presents the TG curves of ZIF-90, PP, and ZIF-90@PP separators. ZIF-90 exhibits no significant thermal decomposition at 300 °C. When the temperature rises to 620 °C, the weight loss rate is only 24%. In contrast, the PP separator decomposes completely at 470 °C, while the ZIF@PP separator shows a weight loss of 49% at the same temperature. These results confirm the improvement of thermostability of the separator when the ZIF-90 functional coating is introduced. The N_2_ adsorption–desorption curve of ZIF-90 is illustrated in Figure 1d. The rapid increase in N_2_ adsorption capacity at low relative pressure is attributed to the existence of micropores in ZIF-90. The small step observed at higher pressure is likely due to slight pore contraction caused by aldehyde groups on the imidazole-2-carboxaldehyd links [44], and the very small hysteresis loop suggests the presence of meso/micropores formed by the packing of crystal particles [46,48,49]. According to the data, the BET surface area of ZIF-90 is 657.86 m^2^·g^−1^, and the average pore diameter is 2.50 nm (inset in Figure 1d).

To observe the apparent morphology of the materials, the ZIF-90 and ZIF-90@PP composite separators were analyzed using SEM. Figure 2a shows that ZIF-90 is a cage-like structure with a regular hexagonal crystal form, and the average particle size is about 500 nm. After coating the right side of the PP separator with ZIF-90 while leaving the left side unprocessed (Figure 2b), it is observed that the untreated left side reveals the irregular pores of the PP separator, whereas the coated right side shows ZIF-90 uniformly covering the PP separator. Notably, there is no evidence of coating detachment after repeatedly bending the ZIF-90@PP separator (inset in Figure 2b), illustrating its excellent flexibility and strong adhesion. Figure 2c reveals that the coating thickness is about 3.87 μm, which is considered moderate. The EDS mapping of the single ZIF-90 particle confirms the uniform distribution of constituent elements of C, O, N, and Zn (Figure 2d).

The fantastic wettability between the separator and the electrolyte can reduce the ion migration resistance, which can be evaluated by measuring their contact angles. Figure 3a, b illustrate that the electrolyte droplets spread rapidly on the surface of the ZIF-90@PP composite separator, while the contact angle formed by the same amount of electrolyte on the blank PP separator is significantly larger. The result demonstrates that the introduction of ZIF-90 can enhance the wettability between the separator and the electrolyte, thereby improving the separator’s ability to retain electrolytes and maintain stable performance during cycling.

### 3.2. Electrochemical Performance

To investigate the potential of the ZIF-90@PP composite separator in improving the electrochemical stability of batteries, Li || Li symmetric cells were assembled using PP and ZIF-90@PP separators. At a specific capacity of 2 mAh cm^−2^ and a current density of 2 mA cm^−2^, the symmetrical cells with the ZIF-90@PP separator demonstrated stable lithium plating/stripping process for over 780 h (Figure 4a). In contrast, the symmetrical cells with the PP separator exhibited a gradual increase in voltage after 150 h, indicating the formation and accumulation of lithium dendrites during cyclic charge and discharge, leading to increased internal resistance within the cells. Furthermore, the inset of Figure 4a also illustrates a flatter voltage plateau and lower overpotential for the symmetric cells with the ZIF-90@PP separator compared to the cells with PP. These findings provide evidence that the desolvation of Li^+^ by polar groups on the ZIF-90 skeleton accelerates the transport of Li^+^ and reduces the reduction of solvent molecules to impurities on the surface of lithium metal anodes, resulting in a more stable SEI (solid electrolyte interface). The higher voltage observed in the first 50 h of Figure 4a is attributed to SEI formation.

The electrochemical impedance spectroscopy (EIS) of Li || LFP full cells utilizing different separators further corroborates the rapid kinetics of lithium deposition facilitated by ZIF-90 (Figure 4b). The semicircle observed in the high-frequency region corresponds to the charge transfer resistance (R_ct_), while the straight line in the low-frequency region is associated with the diffusion of Li^+^. The reduced semicircle diameter of the cells with ZIF-90@PP indicates a significant decrease in R_ct_ post-modification. Based on fitting data, the R_ct_ of the cells with PP and ZIF-90@PP separators are 79.32 Ω and 41.94 Ω, respectively. This reduction in R_ct_ is primarily due to the enhanced wettability of the ZIF-90@PP separator with the electrolyte. Moreover, the diffusion resistance of Li^+^ in the cells based on ZIF-90@PP separators decreases, as evidenced by the increase in the linear slope in the low-frequency region. The decrease in both R_ct_ and diffusion resistance directly proves that ZIF-90 promotes the rapid transfer of Li^+^, thus reducing the probability of the formation of lithium dendrites and dead lithium.

To further evaluate the effect of the ZIF-90@PP separator on the battery’s cycle performance, Li || LFP full cells with different separators were initially activated for three cycles at 0.1 C and then charged and discharged at 0.5 C (1 C = 150 mA g^−1^). The average load per unit area of the ZIF-90@PP separator is 0.10 mg cm^−2^. As shown in Figure 4c, the average coulombic efficiency of the Li || LFP full cells with ZIF-90@PP (98.67%) and PP (98.50%) separators is not significantly different. However, the cyclic stability of the former is effectively enhanced, as evidenced by the data comparison in Figure 4d. For the discharge specific capacity of the first lap, the cell using the ZIF-90@PP separator (150 mAh g^−1^) outperforms the one using the PP separator (148 mAh g^−1^). After 300 cycles, the discharge specific capacity of the former is 107 mAh g^−1^, compared to 74 mAh g^−1^ for the latter.

As shown in Figure 4e, the cells with ZIF-90@PP separators also demonstrate better rate performance. Specifically, they exhibit discharge specific capacities of 156, 141, 135, 119, and 110 mAh g^−1^ at 0.1, 0.5, 1, 3, and 5 C, respectively, whereas the PP-based cells only show the discharge capacity of 139, 132, 126, 110, and 99 mAh g^−1^ at the same rates. The higher discharge specific capacities of the Li || LFP full cells with the ZIF-90@PP separators are attributed to the anion adsorption capacity of ZIF-90, which promotes the dissociation of LiTFSI to accelerate the transport of Li^+^. Notably, the stable SEI and rapid kinetics of lithium deposition also endow the ZIF-90@PP-based cells with higher capacity retention and better rate performance. After 25 cycles, when the current density is reduced to 0.1 C, the discharge specific capacity of the cells using ZIF-90@PP separators can be restored to 154 mAh g^−1^.

### 3.3. Modification Mechanism

To elucidate the influence mechanism of the ZIF-90 on lithium deposition, cells were assembled using stainless steel (SS) to investigate the effect of different separators on the ionic conductivity of the electrolyte. As shown in the EIS of Figure 5a, the ionic conductivity of SS-SS cells based on PP and ZIF-90@PP separators were calculated to be 1.61 and 1.10 mS cm^−1^. The relatively lower ionic conductivity of ZIF-90@PP-based cells may be attributed to the fixation of anions by ZIF-90. To further validate this hypothesis, the Li || Li symmetric cells were assembled using PP and ZIF-90@PP separators. After standing overnight, the polarization current, initial impedance, and steady-state impedance of the cells were measured under a polarization voltage of 10 mV. The migration number of Li^+^ ($t_{{\mathrm{Li}}^{+}}$) was quantitatively calculated by the classical Bruce–Vincent–Evans formula [5,50]: $t_{{\mathrm{Li}}^{+}}=I_{s}\left(\Delta V-I_{0}R_{0}\right)/\left(\Delta V-I_{s}R_{s}\right)$, where $I_{0},\;\;I_{s},\;\;R_{0},\;\;R_{s}$ represent initial polarization current, steady-state polarization current, initial resistance, and steady-state resistance, respectively. Comparing the EIS diagrams in Figure 5b, c, the smaller semicircle diameter of the ZIF-90@PP separator indicates that its interface resistance before and after polarization is lower than that of the PP separator. The $t_{{\mathrm{Li}}^{+}}$ of Li || Li symmetric cells with PP and ZIF-90@PP separators is 0.22 and 0.71, respectively. The $t_{{\mathrm{Li}}^{+}}$ increases by more than three times after modification, indicating that the side reactions of TFSI^−^ decrease and Li^+^ becomes the main carrier in the electrolyte. This result can be attributed to the fixation effect of ZIF-90 on TFSI^−^ and the desolvation effect on Li^+^.

To further determine the role of ZIF-90 on the lithium nucleation barrier, the first cycle nucleation overpotential of Li || Cu half-cells was studied. A smaller nucleation overpotential of the cells reflects a lower energy barrier for lithium deposition [5]. In Figure 5d, the nucleation overpotential of Li || Cu cells with PP and ZIF-90@PP separators at a current density of 2 mA cm^−2^ is 73 mV and 49 mV. Obviously, the ZIF-90 coating effectively reduces the energy barrier of lithium nucleation, which may be due to the fact that the regular channel of ZIF-90 homogenizes the flux of Li^+^ to reduce the local current density [5]. The decrease in the lithium nucleation barrier is beneficial to the uniform deposition of lithium, thereby inhibiting the growth of lithium dendrites.

The surface morphology of the lithium anode of Li || Li symmetric cells after plating/stripping for 180 h at a current density of 2 mA cm^−2^ is shown in Figure 6a,b. The lithium anode of the cells with PP separators appears as rough moss-like dendrites. In contrast, the cells with ZIF-90@PP display a relatively smooth lithium anode surface. It can be seen that ZIF-90@PP separator can indeed inhibit the growth of lithium dendrites. Figure 6c shows the schematic illustration of the effect of ZIF-90 on the LMBs. The nano-channels in ZIF-90, which are more regular than the PP separator, reduce the passage of anions and homogenize the flux of Li^+^. More importantly, the interaction of TFSI^−^ with aldehyde groups acting as Lewis acids, along with the attraction of Li^+^ by nitrogen-containing anionic groups, facilitates the dissociation of LiTFSI to release more Li^+^. This alleviates space charge effects near the electrode surface and accelerates Li^+^ transport. The excellent electrolyte wettability of ZIF-90 also enhances the electrolyte retention capacity of separators. Consequently, the rapid transport and uniform deposition of Li^+^ suppress lithium dendrite growth.

## 4. Conclusions

To sum up, a ZIF-90@PP composite functional separator was successfully designed for LMBs to inhibit the growth of lithium dendrites. ZIF-90 features excellent thermal stability, high specific surface area, and regular nano-pores. The desolvation of Li^+^ and the fixation of anions by ZIF-90 significantly increase $t_{{\mathrm{Li}}^{+}}$ and accelerate the transport of Li^+^. The superior electrolyte retention derived from the electrolyte wettability of ZIF-90 contributes to maintaining stable battery performance during cycling. Moreover, the regular nano-channels of ZIF-90 coating effectively reduce the energy barrier of lithium nucleation, promoting the uniform deposition of Li^+^. As a result, the Li || Li cells with the ZIF-90@PP separator exhibit a stable lithium plating/stripping process over 780 h at 2 mA cm^−2^ without an internal short circuit. The Li || LFP cells with the ZIF-90@PP separator demonstrate improved cycle stability and rate performance. These results indicate that the ZIF-90@PP composite functional separator is expected to enhance the electrochemical performance of LMBs.

## Figures and Tables

**Figure 1 nanomaterials-14-00975-f001:**
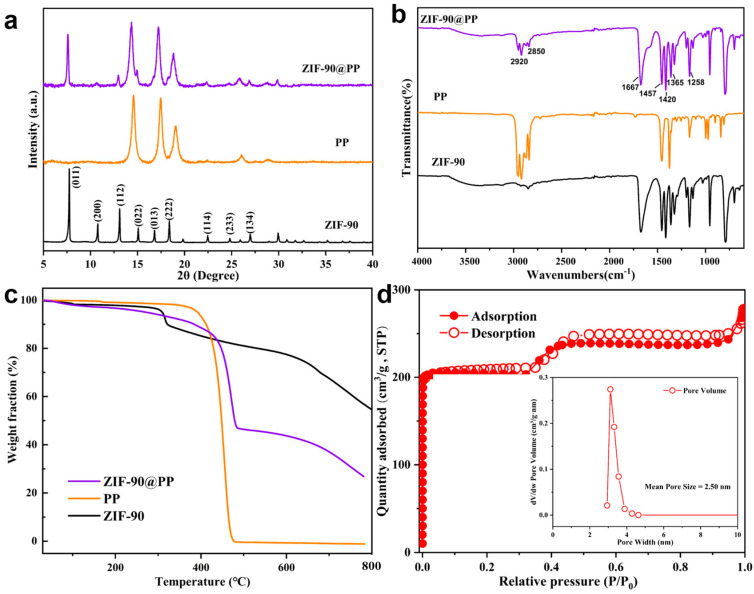
(**a**) XRD patterns, (**b**) ATR–FTIR spectra, and (**c**) TG curves of ZIF-90, PP, and ZIF-90@PP composite separator. (**d**) N_2_ adsorption and desorption isotherm of ZIF-90. The inset shows the pore size distribution of ZIF-90.

**Figure 2 nanomaterials-14-00975-f002:**
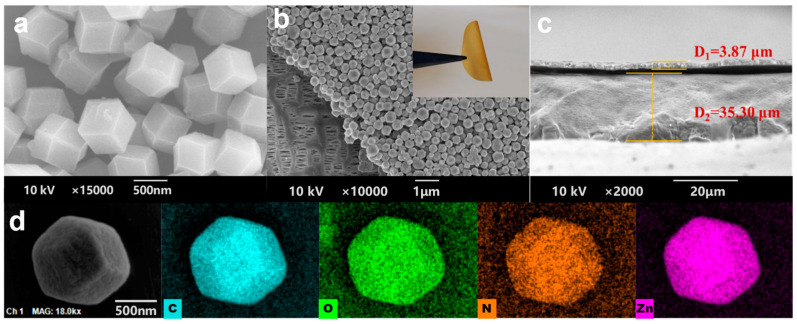
SEM images of the (**a**) ZIF-90, (**b**) surface, and (**c**) section of the ZIF-90@PP composite separator. The inset in (**b**) shows the bent ZIF-90@PP separator. (**d**) SEM image and corresponding EDS maps of the C, O, N, and Zn elements.

**Figure 3 nanomaterials-14-00975-f003:**
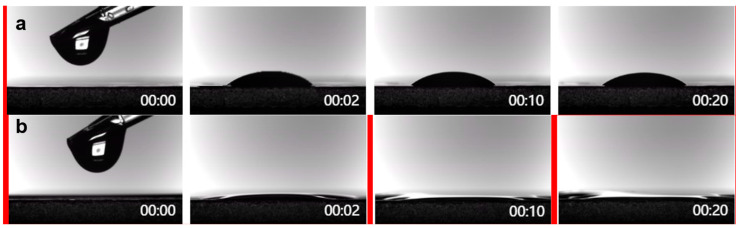
Contact angles of (**a**) PP and (**b**) ZIF-90@PP separator.

**Figure 4 nanomaterials-14-00975-f004:**
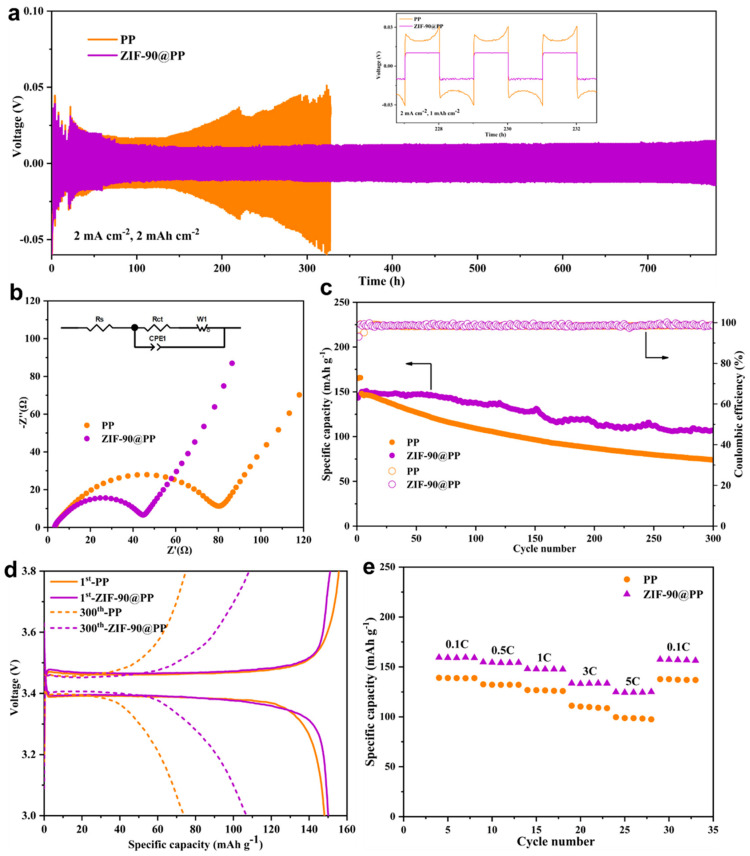
(**a**) Cyclic performance of Li || Li symmetric cells based on PP and ZIF-90@PP separators at a current density of 2 mA cm^−2^. (**b**) Electrochemical impedance spectrum of Li || LFP cells. (**c**) Long cycle performance of Li || LFP cells based on PP and ZIF-90@PP separators with loading of 0.10 mg cm^−2^ at a rate of 0.5 C. (**d**) The 1st and 300th charge–discharge profiles of Li || LFP cells in (**c**). (**e**) Rate performance of Li || LFP cells based on PP and ZIF-90@PP separators.

**Figure 5 nanomaterials-14-00975-f005:**
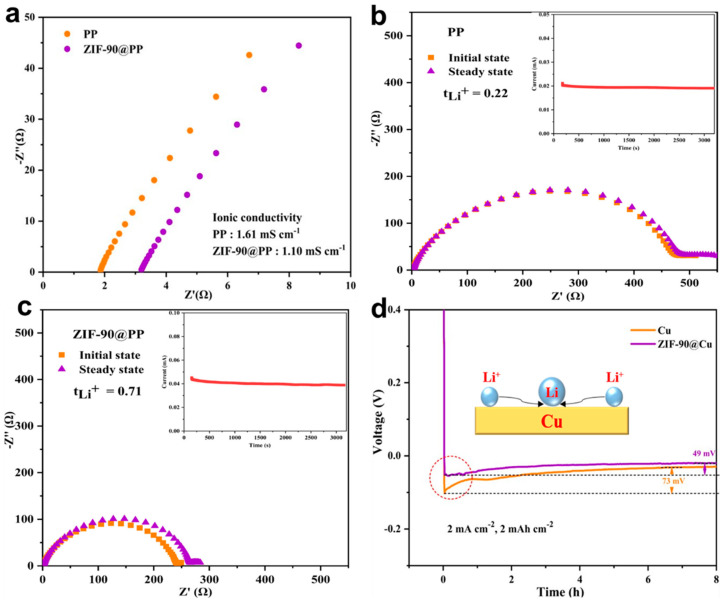
(**a**) Impedance of SS-SS symmetric cells with PP and ZIF-90@PP separators. (**b**) EIS of the Li || Li symmetric cell with PP separator before and after polarization. The inset shows the current–time profile. (**c**) EIS of the Li || Li symmetric cell with ZIF-90@PP separator before and after polarization. The inset shows the current–time profile. (**d**) Nucleation overpotential of Li || Cu half-cells with PP and ZIF-90@PP separators.

**Figure 6 nanomaterials-14-00975-f006:**
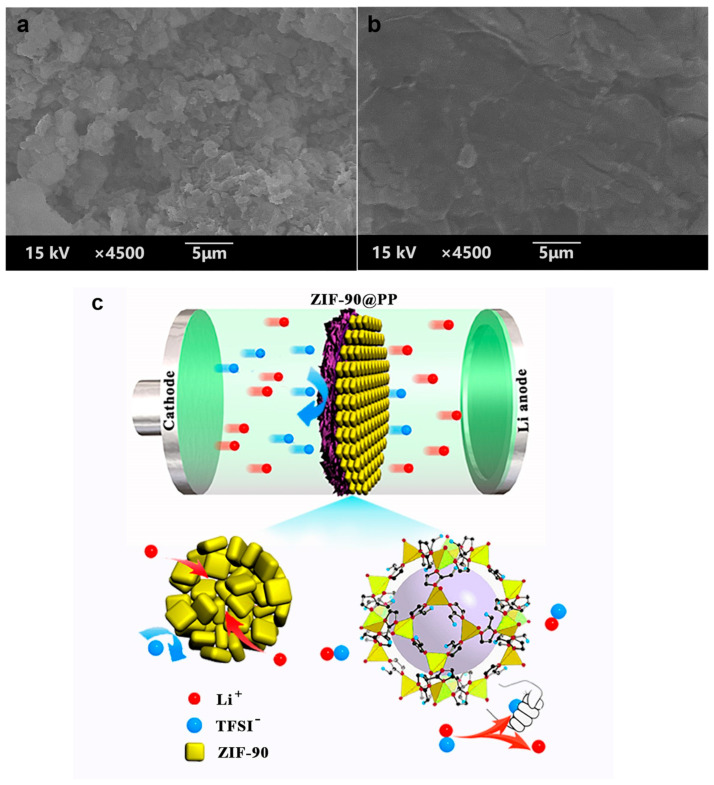
(**a**) SEM image of cycled Li metal with PP separator at 2 mAh cm^−2^ after 180 h. (**b**) SEM image of cycled Li metal with ZIF-90@PP separator at 2 mAh cm^−2^ after 180 h. (**c**) Schematic illustration of the effect of ZIF-90 on the LMBs.

## Data Availability

Data are available within the article.

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
