# Peer review of "Long-Term Stable Cycling of Dendrite-Free Lithium Metal Batteries Using ZIF-90@PP Composite Separator"

_nanomaterials, 2024, doi:10.3390/nano14110975_

Round 1

Reviewer 1 Report

Comments and Suggestions for Authors

The authors present a study on a kind of composite separator (ZIF-90@PP) consisting of zeolite imidazole framework-90 (ZIF-90) and polypropylene (PP) to promote the uniform deposition of Li+ and inhibit the growth of lithium dendrites in lithium-ion batteries.

In my opinion, the manuscript can be impactful for the materials science community working on lithium batteries but needs revision before it can be considered for publication.

Here is a list of my major concerns about the manuscript in its present form:

Q1) The introduction section omits a recent article published by the Meng group which is based on a similar approach to the present manuscript:

Molecules 2024, 29(8), 1759; https://doi.org/10.3390/molecules29081759

The authors must cite and clearly indicate the contribution of the present manuscript with respect to their previous one.

Q2) The manuscript lacks chemical characterization of samples, please include FTIR or Raman spectroscopy on the samples.

Q3) Please, discuss the effect of broadening of XRD peaks associated with the loss of crystallinity of samples. Can authors estimate the crystallite size variations in the process?

Q4) Considering that there are notorious variations in the crystallite size when coated in PP membranes, authors must perform TGA and nitrogen adsorption-desorption analyses for the ZIF-90@PP sample and include further discussion.

Q5) Authors must discuss further the TFSI- trapping effect of ZIF-90 based on literature (experimental or theoretical). Is it just due to the size effect? What about the interactions between imidazolate and TFSI anions? Isn't expected a charge repulsion effect between imidazolate and TFSI anions?

Author Response

Dear Reviewer:

Thank you very much for your comments on our manuscript. In the revised version, we have fully taken into account all the points you raised. All the changes and revised contents can be clearly seen with highlight in the revised manuscript. Please see the attachment. The detailed responses are given as follows:

Question 1: The introduction section omits a recent article published by the Meng group which is based on a similar approach to the present manuscript. Molecules 2024, 29(8), 1759; https://doi.org/10.3390/molecules29081759. The authors must cite and clearly indicate the contribution of the present manuscript with respect to their previous one.

Response: Thanks for your valuable suggestion. The article you mentioned [42] has been cited (line 77-80) and we elucidate the innovations of our manuscript compared to that article.

Question 2: The manuscript lacks chemical characterization of samples, please include FTIR or Raman spectroscopy on the samples.

Response: Thanks for your valuable suggestion. The FTIR spectra of all samples have been provided in the revised manuscript. The results align with the literatures [44,46], confirming that ZIF-90 was successfully synthesized and coated on the PP separator (line 154-163).

Question 3: Please, discuss the effect of broadening of XRD peaks associated with the loss of crystallinity of samples. Can authors estimate the crystallite size variations in the process?

Response: Thanks for your valuable suggestion. As a precautionary measure, we retested the XRD using new instrumentation for several times, and the results show excellent reproducibility. After coating ZIF-90 onto the PP separator, only a slight broadening of the diffraction peaks was observed. Taking the first three strong diffraction peaks of ZIF-90 as an example, we calculated the average size of the crystallites using the Scherrer equation. The data show that the average crystallite size of ZIF-90 decreased from 0.79 nm to 0.74 nm after coating on the PP separator, possibly due to slight disruption of the ZIF-90 crystal structure during ultrasonic dispersion in methanol, yet this change can be disregarded (line 163-164).

Question 4: Considering that there are notorious variations in the crystallite size when coated in PP membranes, authors must perform TGA and nitrogen adsorption-desorption analyses for the ZIF-90@PP sample and include further discussion.

Response: Thanks for your valuable suggestion. The TG analysis for the PP and ZIF-90@PP separator was added (line 167-173). The data confirms the improvement of thermostability of the separator when introduced the ZIF-90 functional coating. Furthermore, the slight variations in crystallite size can be demonstrated through XRD patterns in Figure 1 (a) (line 163-164), so the adsorption-desorption analyses for the ZIF-90@PP is not performed.

Question 5: Authors must discuss further the TFSI- trapping effect of ZIF-90 based on literature (experimental or theoretical). Is it just due to the size effect? What about the interactions between imidazolate and TFSI anions? Isn't expected a charge repulsion effect between imidazolate and TFSI anions?

Response: Thanks for your careful suggestion. The relevant explanations have been added to the manuscript (line 82-84). According to Lewis acid-base theory, TFSI- acts as a Lewis base, while the aldehyde groups in ZIF-90 behave as Lewis acids, giving ZIF-90 the ability to immobilize TFSI- from the electrolyte[33,35,42-43], not merely due to size effects. Additionally, the imidazolium ester contains nitrogen-based negative groups that attract Li+ while repelling TFSI-, thereby further promoting the dissociation of LiTFSI.

We have tried our best to improve the manuscript and made changes by following the experts’ requests and suggestions. We really appreciate your valuable comments and suggestions, and wish that the relevant revisions could meet with the acceptance approval for the publication in Nanomaterials.

Very sincerely yours,

Dr. Yuezhong Meng

Reviewer 2 Report

Comments and Suggestions for Authors

see attachment

Comments on the Quality of English Language

English language should be improved particularly in some parts of the paper that I also indicated in the attached file. However, a general revision of the entire paper is suggested

Author Response

Dear Reviewer:

Thank you very much for your comments on our manuscript. In the revised version, we have fully taken into account all the points you raised. All the changes and revised contents can be clearly seen with highlight in the revised manuscript. Please see the attachment. The detailed responses are given as follows:

Question 1: The authors do not cite a paper (DOI: 10.1021/acssuschemeng.9b03854 ) in which a separator based on PP and ZIF-67 is reported. It should be added in references (maybe removing some other not so important paper) and properly commented, when necessary. The authors should also evidence their novelty with respect to this paper.

Response: Thanks for your valuable suggestion. In the revised manuscript, we cited this literature [25] and elaborated on it (line 67-74). In comparison to ZIF-67 synthesized from cobalt nitrate hexahydrate and 2-methylimidazole, ZIF-90 is synthesized from the cheaper zinc nitrate hexahydrate and imidazole-2-carboxaldehyde containing an aldehyde group. This results in lower production costs, and the aldehyde group acting as a Lewis acid can immobilize TFSI- to release more Li+.

Question 2: I have another important request. The authors try their separator with a not so diffused electrolyte in LIBs but most common in lithium sulphur batteries. Is it in fact most common in LIBs to find LiPF6 or LiClO4 as lithium salts in different carbonates mixtures. They can provide evidence of the well-functioning of this separator also for other electrolytes/salts mixtures? Because I think that if it only functions with a single type of electrolyte, its utility is very limited.

Response: Thanks for your valuable suggestion. LiTFSI electrolyte is more friendly to LMB compared to ester-based electrolytes. We believe LMB is a candidate for next-generation high-energy-density batteries. To explore functional materials suitable for highly stable LMB, we designed the ZIF-90@PP composite separator and chose LiTFSI electrolyte instead of traditional lithium-ion battery electrolytes which are more prone to corrode lithium metal anodes. But your suggestion is highly constructive. Considering that different anodes require matching with different electrolytes, we will attempt to use various electrolytes to assemble full cells of LMB, which is our next work plan.

Question 3: I think that the list of peaks and Miller indices of ZIF-90 in the text is useless because they are also reported in Fig. 1a. Please remove them. When we describe XRD we have pattern not spectrum (line 142). The comments about the XRD part are very limited, they should be expanded. For example, the pattern of PP is not commented and that of ZIF-PP has very large peaks, in some cases due to the overlapping of ZIF and PP peaks, but in other cases, the same peaks of ZIF alone are very broad. Comments and explanations should be mandatory provided.

Response: Thanks for your careful suggestion. We deleted the list of peaks and Miller indices of ZIF-90 in the text and changed "spectrum" to "pattern" (line 150-152). As a precautionary measure, we retested the XRD using new instrumentation for several times, and the results show excellent reproducibility. After coating ZIF-90 onto the PP separator, the slight broadening of the diffraction peaks can be disregarded. We have also made revisions regarding the comments on the XRD pattern. (line 150-154)

Question 4: About the N2 adsorption, I do not agree that the isotherm is of Type I, but it is more similar to a Type IV (in the reference that I cite in the first point is really of Type I, but that of the current paper is different). In addition, the desorption branch is not coincident with that of adsorption at high pressure. The comments about adsorption should be improved.

Response: Thanks for your valuable suggestion. Comments on adsorption have been updated in the revised manuscript. According to more common literatures [44,46,48-49], we provide a more detailed description of the conclusions (line 173-180).

Question 5: Fig. 2b as it is useless, nothing can be seen, as well as Fig. 2c, we cannot see that ZIF-90 covers the separator. Images with high magnification are necessary to clarify this point.

Response: Thanks for your valuable suggestion. The clearer images have been provided in the manuscript (line 191-193). To enhance the comparison, we coated the right side of the PP separator with ZIF-90 and the left side remained unprocessed. The results indicate that the untreated left side reveals the irregular pores of the PP separator, while the coated right side shows ZIF-90 uniformly covering the PP separator.

Question 6: I cannot see a relation between the red box in the SEM image and the EDS maps, because these last ones seem derive from a different SEM image. Please better clarify.

Response: Thanks for your valuable suggestion. To be cautious, we re-scanned ZIF-90 using EDS and provided an explanation in the manuscript (line 190-193).

Question 7: Lines 180-193, improve English.

Response: Thanks for your valuable suggestion. We have improved the English based on your suggestions and the relevant content can be seen in the manuscript (line 204-217).

Question 8: Line 201. The authors say that ZIF-90 with a rich pore structure can accommodate more electrolyte than PP. But they declare that the material was microporous (see adsorption measurements), without a rich pore structure as it is evident from the pore distribution, so I think that it cannot accommodate the electrolyte in the pores. This part should be changed/improved.

Response: Thanks for your valuable suggestion. The original manuscript lacked precision, so we revised the relevant content (line 224-226).

Question 9: In Fig. 6a and 6b nothing can be seen. Please improve these images.

Response: Thanks for your valuable suggestion. We have provided clearer images in the manuscript (line 305-306).

Question 10: Line 277. Again, the authors refer to adsorption of anions, but this should be clarified as I requested in a previous point.

Response: Thanks for your valuable suggestion. According to Lewis acid-base theory, TFSI- acts as a Lewis base, while the aldehyde groups in ZIF-90 behave as Lewis acids, giving ZIF-90 the ability to immobilize TFSI- from the electrolyte. We also added supplementary explanations in the relevant sections of the manuscript. (line 82-84).

We have tried our best to improve the manuscript and made changes by following the experts’ requests and suggestions. We really appreciate your valuable comments and suggestions, and wish that the relevant revisions could meet with the acceptance approval for the publication in Nanomaterials.

Very sincerely yours,

Dr. Yuezhong Meng

Round 2

Reviewer 1 Report

Comments and Suggestions for Authors

The authors have addressed my major concerns about the first version of the manuscript and, in my opinion, it can be accepted for publication.

Author Response

Dear Reviewer:

Thanks very much for your kind work and consideration on publication of our paper.  We would like to express our great appreciation to you.

Thank you and best regards.

Yours sincerely,

Dr Yuezhong Meng

Reviewer 2 Report

Comments and Suggestions for Authors

The authors satisfied the requests. I have some minor points:

- the English should be again revised

- please specify if FT-IR were collected in ATR mode or transmittance

- line 150. Write: the diffraction peaks agree with those reported in the literature for similar materials, confirming....

- line 155. The composite?

- caption of fig. 2. Point (d) write "SEM image and corresponding EDS maps of the C, O, N and Zn elements (not spectra)

Comments on the Quality of English Language

Should be again revised

Author Response

Dear Reviewer:

Thank you very much for your comments for our manuscript. I am sending herewith the revised manuscript (ID: nanomaterials-3021718) of the manuscript entitled “Long-term Stable Cycling of Dendrite-free Lithium Metal Batteries Using ZIF-90@PP Composite Separator”. In the revised version, we have fully taken into account all the points you made. All the changes and revised contents can be clearly seen with highlight in the revised manuscript. Please see the attachment. The detailed responses are given as follows:

Question 1: The English should be again revised.

Response: Thanks for your valuable suggestion. We have made every effort to improve the English, and the revisions can be clearly seen with highlight in the revised manuscript.

Question 2: Please specify if FT-IR were collected in ATR mode or transmittance.

Response: Thanks for your valuable suggestion. The FTIR spectra of all samples were collected in ATR mode, and the test conditions were detailed in the revised manuscript. (line 137-141).

Question 3: line 150. Write: the diffraction peaks agree with those reported in the literature for similar materials, confirming....

Response: Thanks for your careful suggestion. We have revised the manuscript in accordance with your suggestions. (line 153-154).

Question 4: line 155. The composite?

Response: Thanks for your valuable suggestion. This expression lacks precision, so we have replaced "composite" with "characteristic functional groups".(line 158-159).

Question 5: caption of fig. 2. Point (d) write "SEM image and corresponding EDS maps of the C, O, N and Zn elements (not spectra).

Response: Thanks for your careful suggestion. We have revised the manuscript in accordance with your suggestions. (line 196-197).

The suggestion you made is  incredibly helpful to us and we have tried our best to improve the manuscript and made changes by following the experts’ requests and suggestions. We really wish that the relevant revisions could meet with the acceptance approval for the publication in Nanomaterials.

Very sincerely yours,

Dr. Yuezhong Meng